# Investigation of Biomedical Students’ Knowledge on Glaucoma Reveals a Need for Education: A Cross-Sectional Study

**DOI:** 10.3390/healthcare10071241

**Published:** 2022-07-03

**Authors:** Ana Vucinovic, Josipa Bukic, Doris Rusic, Dario Leskur, Ana Seselja Perisin, Tin Cohadzic, Marko Luksic, Darko Modun

**Affiliations:** 1Department of Ophthalmology, University Hospital Centre Split, Spinciceva 1, 21000 Split, Croatia; avucinovic@kbsplit.hr; 2Department of Pharmacy, University of Split School of Medicine, Soltanska 2, 21000 Split, Croatia; drusic@mefst.hr (D.R.); dleskur@mefst.hr (D.L.); aperisin@mefst.hr (A.S.P.); bluksic@mefst.hr (M.L.); dmodun@mefst.hr (D.M.); 3Department of Pediatric Surgery, University Hospital Centre Split, Spinciceva, 21000 Split, Croatia; tcohadzic@kbsplit.hr

**Keywords:** glaucoma, education, knowledge, students, medicine, pharmacy

## Abstract

Background: Glaucoma has been recognized as one of the leading global causes of irreversible blindness. Patients with primary open-angle glaucoma rarely present with visual symptoms, at least early in the course of the disease. It is important to recognize and treat the disease before there are irreversible changes. Methods: This cross-sectional study was conducted at the University of Split School of Medicine from October to November 2021. Participants were biomedical students who completed a questionnaire. Results: In total, 312 students participated in this study. Interestingly, only 12.2% of students identified that primary open-angle glaucoma was asymptomatic. Only 42.6% of all students recognized glaucoma as being the main cause of irreversible blindness. Pharmacy students less frequently recognized high blood pressure and diabetes mellitus as risk factors for glaucoma. Students who completed an ophthalmology course more frequently recognized that successful glaucoma treatment prevents blindness, compared to students who did not complete the course, 79.1% vs. 48.7%, *p* < 0.001. Conclusion: The results showed that students’ knowledge on the subject is low, even after they passed their ophthalmology course. However, knowledge of glaucoma is crucial for early disease identification and the prevention of blindness. Therefore, it is important that all future health care professionals acquire adequate education.

## 1. Introduction

Glaucoma has been recognized as one of the leading global causes of irreversible blindness. Together with other main causes of blindness, cataracts, age-related macular degeneration and diabetic retinopathy, glaucoma is attributed to blindness in more than six million adults over fifty years of age in 2019 [1]. Furthermore, studies have shown that the prevalence of bilateral blindness caused by glaucoma varies from 6% to as much as 16% in Western countries [2]. Glaucoma and other causes contribute to moderate and severe vision impairment in over thirteen million adults. Even though the results of the study by the Blindness and Vision Impairment Collaborators showed a decrease in the prevalence of blindness between 1990 and 2020, there is still an opportunity to reduce the morbidity in this population with early detection and timely intervention [1].

Family history is the main risk factor for glaucoma. However, the associations between glaucoma and several other risk factors were studied in the literature. Different studies considered age, race, medication adverse drug reactions, smoking, hypertension and obstructive sleep apnea as additional factors that contribute to glaucoma incidence [3]. Studies have shown that glaucoma significantly affects patients’ quality of life (QoL). In general, primary open-angle glaucoma patients rarely present with visual complaints, at least early in the course of this disease. However, the disease and also the treatment can have an enormous impact on each patient’s QoL. Even the diagnosis itself can adversely affect the patient’s QoL and trigger anxiety because of the potentially disabling disease [4]. Both Majernikova and Szegedi showed that glaucoma impacts patients’ self-sufficiency, social and work enforcement and social status, which leads to a significantly lower QoL when compared to patients without impairment [5,6]. On a global level, glaucoma also affects health care costs, and results from the study by Frick et al. showed that the annual costs of visually impaired patients included 5.5 billion dollars for medical care and informal care in the United States of America [7].

The global prevalence of glaucoma in the population aged between 40 and 80 years has been estimated at 3.54%. The projection of the glaucoma burden from the study by Tham et al. is that the number of glaucoma patients will increase from 76.0 million worldwide in 2020 to 111.8 million in 2040 [8]. Research on glaucoma has increased in the past decade and glaucoma has been established among the top five most frequently studied eye diseases [9]. However, the increase in glaucoma research has not been followed with an increase in glaucoma awareness in the general population or health care professionals. Moreover, even data on the education of pharmacy, dental and medical students in this field is limited. The results of a study by Martins, conducted in medical students, revealed that 95.1% of the participants considered their level of glaucoma knowledge insufficient. It seems reasonable that proper formal education would result in better understanding of this condition in future health care professionals, and the optimization of glaucoma patient care in their daily practice. Similar findings were observed in a study by Boadi-Kusi et al., which included pharmacy, dental and medical students. Interestingly, these results also revealed that the media had a major role in students’ glaucoma awareness but a limited role in students’ knowledge [10,11,12].

Early glaucoma diagnosis and accurate treatment are the only methods of preventing vision damage in affected patients. Glaucoma screening and public health strategies should be introduced in order to improve outcomes and prevent blindness. In order to raise awareness of the general public on the matter of glaucoma, health care professionals should be knowledgeable on this matter. Biomedical students, as future health care professionals, are exposed to glaucoma training during their formal education. Therefore, the aim of this study was to assess the knowledge and self-practices on glaucoma in biomedical students as future health care professionals.

## 2. Materials and Methods

This cross-sectional study was conducted at University of Split School of Medicine in the academic year 2021/2022. In total, 809 students were enrolled in this academic year and sample size calculation, with confidence level of 95% and margin of error of 5%, showed 261 students were needed for this study. The study was performed from October to November 2021. The included participants were students of pharmacy, medicine and dental medicine, as it was more feasible for the study investigators to approach students in these programs than students in allied health programs offered at another school. The Ethics Committee of the University of Split School of Medicine approved this study, number 2181-198-03-04-21-0034. A consent statement was provided at the beginning of the questionnaire in Google Forms and participation was completely voluntary, with no incentives offered to students. The questionnaire was sent as a link in an emailed invitation to the students’ representatives of each study year of pharmacy, dentistry and medicine. 

The questionnaire used in the study was designed based on previously published questionnaires with some modifications [10,11]. The chosen items from published questionnaires were translated into the Croatian language and after that back translated into the English language by a native English speaker, in order to ensure a validated questionnaire. The final questionnaire consisted of three sections and 23 items. The first section, comprised 8 items, collected demographic data of the students: study program, gender, age, study year, grade earned in their ophthalmology course, family history of glaucoma, students affected with glaucoma and sources of information on glaucoma. The second section questioned students’ knowledge of glaucoma and it comprised 12 items. The students were asked about the most common cause of irreversible blindness, if glaucoma blindness is irreversible, most common type of glaucoma, cause of primary open-angle glaucoma, risk factors for primary open-angle glaucoma, symptoms of primary open-angle glaucoma, main diagnostic test for glaucoma, glaucoma treatment, treatment outcome, pharmacotherapy duration, pharmacotherapy itself and satisfaction with their formal education on glaucoma. The third section consisted of 3 items on students’ self-care practice. Students were asked about their last ophthalmological examination, their attitude towards glaucoma treatment and surgery. The questionnaire is available as Appendix A.

MedCalc ver. 11.5.1.0 statistical software for Windows (MedCalc Software, Ostend, Belgium) was used for data analysis in this study. Data were expressed as whole numbers and percentages for categorical variables. The chi-squared test was used to asses associations. Fisher’s exact test was used when proportion of answers was 5% or less or there were only 5 or fewer answers. This test was conducted using SPSS (V.16.0, IBM). The statistical significance was defined as *p* < 0.05.

## 3. Results

In total, 312 students participated in this study. Demographic characteristics of the study participants are presented in Table 1. The number of students from the pharmacy and medicine program was similar, whereas there were fewer dental medicine students. Moreover, almost 77% of study participants were female. One third of the participants were first year students and 7% of students had a family member with glaucoma.

Dental and medical students selected their educational materials most commonly, while the majority of pharmacy students selected the internet, as their source of information on glaucoma. Only 42.6% of all students and 53.5% of students who completed the ophthalmology course recognized glaucoma as being the main cause of irreversible blindness. Moreover, even fewer students, 38.8% overall and 48.8% of students who completed the course, recognized that glaucoma causes irreversible blindness. The difference was observed in this knowledge item between study programs as 49.1% of medical students had knowledge on irreversible blindness as a consequence of glaucoma, compared to 31.5% of pharmacy and 35.7% of dental students, *p* = 0.016.

The majority of all students stated they did not know the most common type of glaucoma, 61.9% of them. In total, 28.8% of students recognized primary open-angle glaucoma as the most common glaucoma type. Medical students most frequently reported primary open-angle glaucoma, 39.3% of them, compared to 20.8% of pharmacy and 27.1% of dental students, *p* = 0.006. Again, 48.8% of students who completed the ophthalmology course recognized the most common glaucoma type.

Genetic factors as the cause of open-angle primary glaucoma were acknowledged by only 11.9% of all students, and again medical students most frequently (20.7%) reported this factor compared to 6.2% of pharmacy and 8.6% of dental students, *p* < 0.001. A difference was observed between students who completed the ophthalmology course and other students, 22.4% and 8.0%, respectively, *p* < 0.001. Students’ knowledge of the main risk factors for primary open-angle glaucoma is presented in Table 2. The majority of students recognized elevated intraocular pressure as a risk factor. However, students were less likely to recognize other risk factors, especially family history, corticosteroid use and black race. Pharmacy students recognized high blood pressure, black race and corticosteroid use the least, compared to dental and medical students.

Only 12.2% of all students correctly identified that primary open-angle glaucoma was asymptomatic. Predominantly, those students were from medical studies (47.4%) followed by pharmacy (31.6%) and dental (21.1%) studies. A statistically significant difference was observed in 60.5% of students who completed the course and identified asymptomatic disease compared to 39.5% of students who did not complete this course, *p* < 0.001. Furthermore, 16.3% and 3.8% of all students thought that examination of visual acuity and magnetic resonance imaging were among the main diagnostic tests for glaucoma. Thirty-two percent of all students correctly recognized that a fundus examination is an important test in the assessment of glaucoma. A larger proportion (54.7%) of students who completed the course recognized the fundus examination, compared to 23.5% of other students, *p* < 0.001. Moreover, the difference in the recognition of the fundus examination was observed in 44.0% of medical students compared to 29.0% of dental and 27.0% of pharmacy students. A large proportion (65.1%) of all students identified tonometry as an important glaucoma diagnostic test and only 29.8% of students identified perimetry. Of the students who completed the course, larger proportions identified tonometry (87.2%) and perimetry (50.0%) compared to 56.6% and 22.1% of students who did not complete the ophthalmology course, *p* < 0.001 for both tests.

Students’ knowledge of glaucoma treatment is presented in Figure 1. Medical students most commonly recognized surgery as an option for glaucoma treatment, compared to pharmacy and dental students, *p* < 0.001. A larger proportion of students who completed the ophthalmology course recognized pharmacotherapy (66.3% vs. 45.6%, *p* < 0.001), surgery (65.1% vs. 41.6%, *p* < 0.001) and laser therapy (55.8% vs. 20.8%, *p* < 0.001) as treatments, compared to students who were not enrolled on the ophthalmology course.

In total, 57.1% of all students answered correctly that successful glaucoma treatment promotes disease control and the prevention of blindness. This knowledge was more common among 79.1% of the students who completed the ophthalmology course, compared to 48.7% of the other students, *p* < 0.001. Furthermore, only 42.8% of all students knew that glaucoma patients would need to use pharmacotherapy during their lifetime. This knowledge was more common in 61.6% of students who completed the course compared to 35.6% of those who did not complete the course, *p* < 0.001. Only 38.8% of pharmacy students had this knowledge, compared to 42.9% of dental and 47.3% of medical students. However, pharmacy students were least likely to state they did not know any drugs available for glaucoma treatment, as 66.2% of pharmacy students did not know any drugs, compared to 90.0% of dental and 78.6% of medical students. A minority (15.6%) of all students replied that the knowledge offered by their study curricula is sufficient for caring for glaucoma patients in their future professions. A difference between the students who completed the ophthalmology course and their colleagues was not observed.

The majority (46.9%) of students had visited an ophthalmologist in the past year. Furthermore, most commonly, 76.5% of students stated that if they were diagnosed with glaucoma they would visit their ophthalmologist regularly and take the suggested treatments consistently. In the case where surgery was the only treatment option available, 82.8% of students would promptly go ahead with the surgery. No difference in the self-care practice items was observed between the study programs.

Additional analyses of two groups, students who completed the ophthalmology course and students who did not complete the ophthalmology course, showed differences between these groups. Firstly, 53.5% of students who completed the course had knowledge that glaucoma is the most common cause of blindness, compared to 38.5% of students who did not complete the course, *p* < 0.001. Furthermore, 48.8% of students who completed the course knew glaucoma-related blindness was irreversible, compared to 35.0% of their colleagues who did not complete the course, *p* < 0.001. A difference was also observed in their knowledge of glaucoma risk factors, such as intraocular pressure, 88.4% vs. 72.6%, respectively, *p* = 0.005, and hypertension, 55.8% vs. 31.0%, respectively, *p* < 0.001. All the students who completed the ophthalmology course recognized family history as a risk factor, compared to 90.7% of their colleagues, *p* = 0.007.

Our results reflect differences in education among the three study programs included in this research. Medical students most commonly recognized surgery as a glaucoma treatment, and they will be the health care practitioners involved in surgery treatment in their future practice. However, they had a lower level of knowledge on glaucoma pharmacotherapy in our study. It is important that medical students, but also dental students, have an appropriate level of knowledge of pharmacotherapy, as they both are health care professionals who can prescribe medication in Croatia. It is also important that all students have an appropriate level of education on glaucoma risk factors because the general population will not always have access to an ophthalmology specialist, especially in rural areas of the country.

## 4. Discussion

Students’ levels of knowledge on glaucoma found in this study are similar to the results of the study conducted by Boadi-Kusi et al. in Ghana. Their results also showed that pharmacy, dental and medical students showed low levels of knowledge on glaucoma. Boadi-Kusi et al. also found a similar proportion (86.8%) of students had knowledge of the association between intraocular pressure and glaucoma. Furthermore, 37.4% of students in their study recognized that glaucoma requires lifelong treatment [12]. Poor glaucoma knowledge in the population of biomedical students was also reported in a study by Martins et al. In this study, a similar proportion of 22.5% of students recognized the genetic origin of primary open-angle glaucoma. However, in the same study, two-thirds of students knew that glaucoma causes irreversible blindness, and in our study this was only 38.8% of students [11]. Generally, the results of our study showed that students’ knowledge on the subject is lacking, even after they passed their ophthalmology course. The underlying reasons most probably lie in the fact that the ophthalmology course is rather short and focused on a great number of subjects. However, knowledge of glaucoma is crucial for early disease identification and the prevention of blindness. Therefore, it is important that all future medical professionals are adequately educated on this subject.

Moreover, glaucoma is not the only eye condition with a high prevalence in the general population. Hence, students should be offered an education focusing on the eye diseases that every health care professional needs to know, especially diabetic retinopathy, hypertensive retinopathy, age-related macular degeneration, etc. This course could be offered at first as an elective course available to all study programs, and students from all three study programs would be enrolled on the same course, which could also encourage interprofessional collaboration in their future practice.

Previous studies show that the general population’s knowledge on glaucoma is inadequate. The results of a study by Alemu et al. showed that better glaucoma awareness by patients was related to more frequent eye examinations [13,14,15]. Pharmacists, as the most accessible health care professionals, have a unique opportunity to raise patients’ awareness of glaucoma and question if their patients schedule their eye examinations as needed. Therefore, it seems reasonable that pharmacy students, who presented with low knowledge on glaucoma in this study, should be offered additional education promptly. Other than their role in raising glaucoma awareness, pharmacists also have an opportunity to improve patients’ adherence to eye drop use [16,17]. Furthermore, pharmacy students will encounter patients with chronic diseases such as high blood pressure and diabetes mellitus in their daily practice. Thus, it is important for pharmacy students to possess knowledge about chronic conditions as being risk factors for glaucoma.

Along with the growth in the investigation of new approaches to glaucoma treatments, such as neuroprotection, electrical stimulation or cell transplantation treatments, the gap in biomedical students’ knowledge about glaucoma treatments will also rise [18,19]. Therefore, future studies should involve the development of new courses on glaucoma for all biomedical students. These courses could be offered as elective courses to students but could also be adapted for practicing health care professionals as a part of their continuing education. This education should raise awareness of glaucoma in students or health care professionals and provide a grounding in appropriate lifestyle modifications for glaucoma patients. This is especially important as research has shown that modifying diet and exercise could have a therapeutic benefit in glaucoma patients [20].

This study had some limitations that can be taken into account while interpreting our results. One of the main limitations was that study participants were biomedical students from a single university in Croatia. In Croatia, the biomedical study field and study programs are medicine, dental medicine, pharmacy and veterinary medicine. A broader sample, including students from other universities and other countries would have demonstrated a need for glaucoma education more clearly. However, as research on this matter is limited, this study provides important information for educators and future health care professionals.

## 5. Conclusions

The results of our study showed that students’ knowledge on the subject is low, even after they passed their ophthalmology course. However, knowledge of glaucoma is crucial for early disease identification and the prevention of blindness. Therefore, it is important that all future health care professionals acquire education on glaucoma and its treatment options. Future studies should involve the development of glaucoma-specific educational interventions for all biomedical students.

## Figures and Tables

**Figure 1 healthcare-10-01241-f001:**
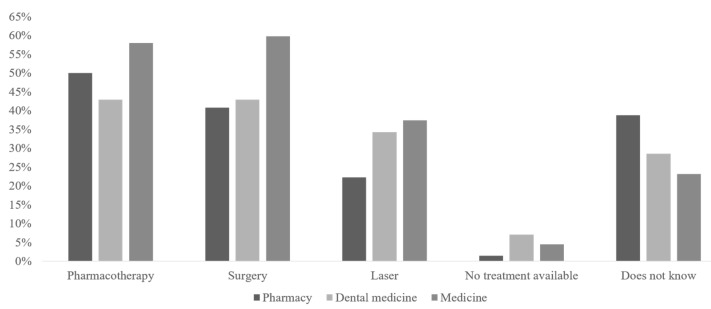
Students’ knowledge of glaucoma treatment. N (pharmacy) = 130. N (dental medicine) = 70. N (medicine) = 112.

**Table 1 healthcare-10-01241-t001:** Demographic data of study participants.

Characteristic	N (%)
Study program	
Pharmacy	130 (41.7)
Dental medicine	70 (22.4)
Medicine	112 (35.9)
Age	
18	19 (6.2)
19	57 (18.5)
20	32 (10.4)
21	33 (10.7)
22	39 (12.7)
23	53 (17.2)
24	44 (14.3)
25	18 (5.8)
26	17 (5.2)
Gender	
Female	240 (76.9)
Male	72 (23.1)
Study year	
First	101 (32.5)
Second	20 (6.4)
Third	40 (12.9)
Fourth	54 (17.4)
Fifth	70 (22.5)
Sixth	26 (8.4)
Completed ophthalmology course	86 (27.6)
Family member with glaucoma	22 (7.1)
Participant with glaucoma	3 (1.0)

**Table 2 healthcare-10-01241-t002:** Students’ knowledge of glaucoma risk factors.

Risk Factor	Pharmacy N (%)	Dental Medicine N (%)	Medicine N (%)	*p*-Value *
**Elevated intraocular pressure**	93 (71.5)	56 (80.0)	91 (81.2)	0.159
**High blood pressure**	37 (28.5)	29 (41.4)	52 (46.4)	0.012
**Diabetes mellitus**	67 (51.5)	36 (51.4)	66 (58.9)	0.450
Neurologic disease	18 (13.8)	9 (12.9)	27 (24.1)	0.058
Younger age	0 (0)	3 (4.3)	0 (0)	0.011 ^#^
Hyperopia	5 (3.8)	4 (5.7)	8 (7.1)	0.508 ^#^
**Myopia**	17 (13.1)	10 (14.3)	28 (25.0)	0.056
**Family history**	11 (8.5)	2 (2.9)	8 (7.1)	0.306 ^#^
Eye trauma	88 (67.7)	52 (74.3)	86 (76.8)	0.266
**Corticosteroid use**	16 (12.3)	10 (14.3)	28 (25.0)	0.025
**Black race**	5 (3.8)	6 (8.6)	8 (7.1)	0.319 ^#^

* Chi-squared test. ^#^ Fisher’s exact. Glaucoma risk factors are bolded.

## Data Availability

All data are presented in the manuscript.

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
