# Peer review of "Investigation of Biomedical Students’ Knowledge on Glaucoma Reveals a Need for Education: A Cross-Sectional Study"

_healthcare, 2022, doi:10.3390/healthcare10071241_

Round 1

Reviewer 1 Report

The study presented here is similar to the other as mentioned from the authors (Saulo Costa Martins 2014 and Samuel B Boadi-Kusi 2014). I would suggest as follow:

·       - to add the questionnaire used for the study as a supplement information;

·       - increase the font size of the figure 1 and

·      - to include the age of the students involved in the study and try to analyse a possible correlation between their age and the knowledge about the glaucoma.

Author Response

The study presented here is similar to the other as mentioned from the authors (Saulo Costa Martins 2014 and Samuel B Boadi-Kusi 2014). I would suggest as follow:

Q:  to add the questionnaire used for the study as a supplement information;

A: Dear reviewer, thank you for this important observation. We have added the questionnaire as a supplementary material.

Q: increase the font size of the figure 1 and

A: Thank you for this important input, we have increased the font to 14 in Figure 1.

Q:  to include the age of the students involved in the study and try to analyse a possible correlation between their age and the knowledge about the glaucoma.

A: We are grateful for this comment. We have added the age to the Table 1 in line 133 of the manuscript. Moreover, we have analysed the possible correlation between students age and the glaucoma knowledge. Unfortunately, the significant correlation between their age and glaucoma knowledge was not observed.

Reviewer 2 Report

The Authors performed a good job analyzing the student knowledge on glaucoma. This work was interesting to focusing the role of ophthalmology course at University on public health. 

Author Response

Q: The Authors performed a good job analyzing the student knowledge on glaucoma. This work was interesting to focusing the role of ophthalmology course at University on public health. 

A: Dear reviewer, thank you for this kind comments.

Reviewer 3 Report

Vucinovic et al. presented an interesting and novel cross-sectional study about students’ knowledge on glaucoma. Nevertheless, the results are influenced by the fact that participants have been enrolled from a single university; in addition, 161 of 312 students were attending the first three years of their course, so it’s comprehensible that their knowledge about glaucoma is weak. For this reason, it might be necessary to better highlight the results of the students that have completed the ophthalmology course.

Moreover, it would be desirable to divide the results according to the specific skills of the three students’ category (pharmacy, medicine, dental medicine), meanwhile clarifying the dental students’ role in glaucoma prevention and treatment.

Finally, at lines 220-221 the authors say that “the ophthalmology course is rather short and focused on a great number of subjects”, hence in the discussion, rather than thinking about a specific glaucoma course,  it might be considered the possibility of identifying the critical issues of the ophthalmology course and focusing the course on the eye disease that every health care professional needs to know (besides glaucoma, also diabetic retinopathy, hypertensive retinopathy, age related macular degeneration…) .

Author Response

Q: Vucinovic et al. presented an interesting and novel cross-sectional study about students’ knowledge on glaucoma. Nevertheless, the results are influenced by the fact that participants have been enrolled from a single university; in addition, 161 of 312 students were attending the first three years of their course, so it’s comprehensible that their knowledge about glaucoma is weak. For this reason, it might be necessary to better highlight the results of the students that have completed the ophthalmology course.

A: Dear reviewer, thank you for your kind words for our study. Indeed, we believe this is an interesting and novel cross-sectional study as it was hard to find literature for our manuscript, especially in high quality journals. Therefore, we honestly believe our results will add to the body of literature in this field. We have added a paragraph in line 216 of the results section, in order to better highlight the results of students that have completed the ophthalmology course:

“Additional analyses of two groups, students who completed ophthalmology course and students who did not complete their ophthalmology course, showed differences between these groups. Firstly, 53.5% of students who completed the course had knowledge that glaucoma is the most common cause of blindness, compared to 38.5% of students who did not complete the course, P<0.001. Further, 48.8% of students who completed the course knew glaucoma related blindness was irreversible, compared to 35.0% of their colleagues who did not complete the course, P<0.001. Difference was also observed in their knowledge of glaucoma risk factors, intraocular pressure 88.4% vs 72.6% P=0.005 and hypertension 55.8% vs 31.0%, P<0.001. All the students who completed the ophthalmology course recognized family history as risk factor, compared to 90.7% of their colleagues, P=0.007.”

Q: Moreover, it would be desirable to divide the results according to the specific skills of the three students’ category (pharmacy, medicine, dental medicine), meanwhile clarifying the dental students’ role in glaucoma prevention and treatment.

A: Thank you for this comment. We have changed the manuscript in line 227 accordingly:

“Our results reflect differences in education among three study programs included in this research. Medical students most commonly recognized surgery as glaucoma treatment, and they will be the health care practitioners involved in surgery treatment in their future practice. However, they had lower level of knowledge on glaucoma pharmacotherapy in our study. It is important that medical students, but also dental students, have appropriate level of knowledge of pharmacotherapy, as they are both the health care professionals who can prescribe medication in Croatia. It is also important that all students have appropriate level of education on glaucoma risk factors, because general population will not always have access to ophthalmology specialist, especially in rural areas of the country.”

Q: Finally, at lines 220-221 the authors say that “the ophthalmology course is rather short and focused on a great number of subjects”, hence in the discussion, rather than thinking about a specific glaucoma course,  it might be considered the possibility of identifying the critical issues of the ophthalmology course and focusing the course on the eye disease that every health care professional needs to know (besides glaucoma, also diabetic retinopathy, hypertensive retinopathy, age related macular degeneration…) .

A: Thank you for this important observation. The discussion section has been expanded accordingly in line 256.

“Moreover, glaucoma is not the only eye condition with high prevalence in general population. Hence, students should be offered an education focusing on the eye dis-eases that every health care professional needs to know, especially diabetic retinopathythy, hypertensive retinopathy, age related macular degeneration etc. This course could be offered at first as an elective course available to all study programs, and students from all three study programs would be enrolled in the same course which could also encourage interprofessional collaboration in their future practice.”

Reviewer 4 Report

Introduction

The introduction provided ample background info on glaucoma itself but does not emphasize the research gap. This section has to be re-written highlighting the need for biomedical students to undergo this training. 

Methods 

The study has obtained the necessary ethical approval and consent from participants. 

Participants were pharmacy, medicine and dental? None from the biomedical program? I think we have to very clear with a term "biomedical" here. This is a course on its own, not to be confused with medical & pharmacy courses. It is also strange to have dental students here. 

No justification on the sample size was provided. 

"chi sq was used to compare differences between.." - what differences do you mean? also, chi sq is a test of association, not differences. 

The authors mentioned two different statistical software used. Why? 

Results

Table 2 - some variables appeared bold. why? 

Discussion should emphasize more on strategies that can be taken or have been taken to improve the knowledge. 

Author Response

Introduction

Q: The introduction provided ample background info on glaucoma itself but does not emphasize the research gap. This section has to be re-written highlighting the need for biomedical students to undergo this training. 

A: We are grateful for this comment, and the introduction section has been changed accordingly in line 72:

“However, increase in glaucoma research has not been followed with an increase in glaucoma awareness in general population. Moreover, even data on education of pharmacy, dental and medical students in this field is limited. It seems reasonable that proper formal education would result in better understanding of this condition in future health care professionals, and optimization of glaucoma patient care in their daily practice.”

Q: Methods 

The study has obtained the necessary ethical approval and consent from participants. 

Participants were pharmacy, medicine and dental? None from the biomedical program? I think we have to very clear with a term "biomedical" here. This is a course on its own, not to be confused with medical & pharmacy courses. It is also strange to have dental students here. 

A: Dear reviewer, thank you for your valuable input. In Croatia, biomedical study field and study programs are medicine, dental medicine, pharmacy and veterinary medicine. This sentence has been added to the limitation section in line 288.

“In Croatia, biomedical study field and study programs are medicine, dental medicine, pharmacy and veterinary medicine.”

We have included dental students as they are also health care professionals which should be aware of the glaucoma risks. In rural areas, dental doctors will sometimes be most available to general public and there is a need to involve them as health professionals in major public health care initiatives. Furthermore, dental doctors in Croatia can prescribe all the prescription medications, just like general practitioners. Therefore, it seems reasonable that they are aware of the drug interactions and contraindications, which also includes medication that can cause glaucoma worsening.

Q: No justification on the sample size was provided. 

A: We apologize for this mistake. Indeed, we did a sample size calculation during the study and freely available calculator results showed that we needed a total of 261 students. The manuscript has been changed accordingly in line 81:

“In total, 809 students were in enrolled in this academic year and sample size calculation, with confidence level of 95% and margin of error of 5% showed 261 students were needed for this study.”

Q: "chi sq was used to compare differences between." - what differences do you mean? also, chi sq is a test of association, not differences. 

A: We aimed to find difference in proportions, between survey items (yes/no) and study program. The sentence in methods has been rewritten in line 118;

The chi-squared test was used to asses associations.

Q: The authors mentioned two different statistical software used. Why? 

A: Two manuscript authors analysed data, JB uses MedCalc and DL SPSS in their routine analyses, that is the reason that two software were used.

Results

Q: Table 2 - some variables appeared bold. why? 

A: Correct answers were bolded. We have added this information under the table in line 164. Thank you for this important input.

“Glaucoma risk factors are bolded”

Q: Discussion should emphasize more on strategies that can be taken or have been taken to improve the knowledge. 

A: Thank you for this valuable comment. Literature on the strategies that have been taken in this field is limited, and we believe our study highlights the need of glaucoma awareness and proper formal education in biomedical students. We have added a paragraph to the discussion section in line 256:

“Moreover, glaucoma is not the only eye condition with high prevalence in general population. Hence, students should be offered an education focusing on the eye diseases that every health care professional needs to know, especially diabetic retinopathy, hypertensive retinopathy, age related macular degeneration etc. This course could be offered at first as an elective course available to all study programs, and students from all three study programs would be enrolled in the same course which could also encourage interprofessional collaboration in their future practice.”

Reviewer 5 Report

The results of present study showed that student's knowledge on the glaucoma is low, even after they passed their ophthalmology course. However, knowledge of glaucoma is critical for early disease identification and prevention of blindness. Therefore, it is important that all future health care professionals acquire education on glaucoma and its treatment options.

The manuscript was well written and the data is interesting. I have a minor concern.

Figure 1, the student number of pharmacy, dental, medical students should be listed in the figure legend.

Author Response

Q: The results of present study showed that student's knowledge on the glaucoma is low, even after they passed their ophthalmology course. However, knowledge of glaucoma is critical for early disease identification and prevention of blindness. Therefore, it is important that all future health care professionals acquire education on glaucoma and its treatment options.

The manuscript was well written and the data is interesting. I have a minor concern.

A: Dear reviewer, thank you for recognizing the importance of our manuscript.

Q: Figure 1, the student number of pharmacy, dental, medical students should be listed in the figure legend.

A: We are grateful for this important observation. The following text has been added to line 192 in the legend of the Figure 1:

“N (pharmacy)=130

N (dental medicine)=70

N (medicine)=112”

Round 2

Reviewer 1 Report

In the present form I think that the paper could be accepted for publication.

Author Response

Dear reviewer, thank you for your recommendation to accept our manuscript.

Reviewer 4 Report

I appreciate the work done to amend the manuscript. I have a few concerns that I hope the authors can address before the manuscript can be considered for acceptance. 

Introduction - authors added a paragraph to justify the research gap. However, this raises another question on "what is the limited research done among these students?" - line 74/75. Give a couple of examples. 

Statistical analysis - Ideally analyses are to be conducted using one software to ensure uniformity. Please work on this. 

Table 1 - age should have been given as median or mean, depending on normality

Author Response

I appreciate the work done to amend the manuscript. I have a few concerns that I hope the authors can address before the manuscript can be considered for acceptance. 

Q: Introduction - authors added a paragraph to justify the research gap. However, this raises another question on "what is the limited research done among these students?" - line 74/75. Give a couple of examples. 

A: Thank you for this important comment. We have added a text in line 74 of the manuscript:

"Results of a study by Martins, conducted in medical students, revealed that 95.1% of the participants considered their level of glaucoma knowledge insufficient. It seems reasonable that proper formal education would result in better understanding of this condition in future health care professionals, and optimization of glaucoma patient care in their daily practice. Similar findings were observed in study by Boadi-Kusi et al., which included pharmacy, dental and medical students. Interestingly, these results also revealed that the media had a major role in students’ glaucoma awareness, but a limited role in students’ knowledge [10-12]."

Q: Statistical analysis - Ideally analyses are to be conducted using one software to ensure uniformity. Please work on this. 

A; We are grateful for this advice and plan to use only one software in our future studies.

Q: Table 1 - age should have been given as median or mean, depending on normality

A: The age incorporation in Table was recommended by Reviewer 1 in this form.